

# A systematic review on literature-based discovery workflow

Menasha Thilakaratne, Katrina Falkner and Thushari Atapattu

Faculty of Engineering, Computer and Mathematical Sciences, The University of Adelaide, Adelaide, South Australia, Australia

## ABSTRACT

As scientific publication rates increase, knowledge acquisition and the research development process have become more complex and time-consuming. Literature-Based Discovery (LBD), supporting automated knowledge discovery, helps facilitate this process by eliciting novel knowledge by analysing existing scientific literature. This systematic review provides a comprehensive overview of the LBD workflow by answering nine research questions related to the major components of the LBD workflow (i.e., input, process, output, and evaluation). With regards to the *input* component, we discuss the data types and data sources used in the literature. The *process* component presents filtering techniques, ranking/thresholding techniques, domains, generalisability levels, and resources. Subsequently, the *output* component focuses on the visualisation techniques used in LBD discipline. As for the *evaluation* component, we outline the evaluation techniques, their generalisability, and the quantitative measures used to validate results. To conclude, we summarise the findings of the review for each component by highlighting the possible future research directions.

## INTRODUCTION

Due to the exponential growth of scientific publications, keeping track of all research advances in the scientific literature has become almost impossible for a scientist (*Cheadle et al., 2017*). As a result, scientific literature has become fragmented and individual scientists tend to deal with fragments of knowledge based on their specialisation. Consequently, valuable implicit associations that connect these knowledge fragments tend to remain unnoticed since scientists in each specialisation have only seen part of the big picture. Literature-Based Discovery (LBD) supports cross-disciplinary knowledge discovery to elicit these hidden associations to recommend new scientific knowledge. The recommended novel associations can greatly assist scientists in formulating and evaluating novel research hypotheses (*Ganiz, Pottenger & Janneck, 2005*). While reducing the time and effort, this will also promote scientists to discover new areas of research.

### Brief history

LBD was developed as a research field from the medical discoveries published by Swanson since 1986. In his first discovery, he manually analysed the titles of two literature; *Fish*

Corresponding author
Menasha Thilakaratne,
menasha.thilakaratne@adelaide.edu.au

*oil* and *Raynaud's disease* (*Swanson, 1986*). Swanson has observed that the patients with Raynaud tend to have high *blood viscosity* and high *platelet aggregation*. He has also noted that fish oil contains EPA (eicosapentaenoic acid) that helps to decrease the *blood viscosity* and *platelet aggregation*. By combining these knowledge pairs, he generated the hypothesis; *"Raynaud can be cured using fish oil"*. Furthermore, he also observed that the two literature he was referring are *disjoint*, i.e., the articles in the two literature sets have not mentioned, cited or co-cited each other. Consequently, he published these findings that were deduced using the ABC model (see *Discovery Models* section). His second discovery followed the same process where he manually examined the titles of *Migraine* and *Magnesium* to detect implicit associations that connects the two literature (*Swanson, 1988*). Later, his observations were proven through laboratory experiments that demonstrate the validity of his thinking process (*Ramadan et al., 1989*).

Even though the early work of Swanson was mostly performed manually by merely analysing the article titles and their word co-occurrence frequencies, they formed the foundation of the field. In accordance with Swanson's experiments, the existing disperse knowledge fragments in literature can be accumulated in such a way to develop novel semantic relationships that have not drawn any awareness before (a.k.a *undiscovered public knowledge*) (*Swanson, 1986*). These connectable disperse knowledge fragments in the literature may exist as; (1) hidden refutations or qualifications, (2) undrawn conclusion from different knowledge branches, (3) cumulative weak tests, (4) analogous problems, and/or (5) hidden correlations (*Davies, 1989*). In a later study, Swanson also pointed out the importance of studying cases where the interaction of the two literature sets is not null (i.e., the literature sets are not disjoint), but populated by few articles (a.k.a *literature-based resurrection* (*Swanson, 2011*), *scientific arbitrage* (*Smalheiser, 2012*)).

## Discovery models

Most of the LBD literature is based on the fundamental premise introduced by Swanson, namely the ABC model (*Swanson, 1986*). It employs a simple syllogism to identify the potential knowledge associations (a.k.a. *transitive inference*). That is, given two concepts *A* and *C* in two disjoint scientific literature, if concept *A* is associated with concept *B*, and the same concept *B* is associated with concept *C*, the model deduces that the concept *A* is associated with the concept *C*. The popular ABC model has two variants named as *open discovery* and *closed discovery*.

Open discovery is generally used when there is a single problem with limited knowledge about what concepts can be involved. The process starts with an initial concept related to the selected research question/problem (A-concept). Afterwards, the LBD process seeks the relevant concepts that ultimately lead to implicit associations (C-concepts). In other words, only the concept A is known in advance and concepts B and C are identified by the LBD process. Therefore, this model can be viewed as a knowledge discovery process that assists to generate novel research hypotheses by examining the existing literature. Unlike the open discovery process, closed discovery model attempts to discover novel implicit associations between the initially mentioned A-concept and C-concept (a.k.a *concept bridges*). Thus it represents hypotheses testing and validation process. More explicitly, the LBD process

starts with user-defined A-concept and C-concept and the output will be the intermediate B-concepts that represents the associations between the two user-defined domains.

Even though the prevalent ABC model have contributed in numerous ways to detect new knowledge, it is merely one of several different types of discovery models that facilitates LBD process. In this regard, *Smalheiser (2012)* points out the importance of thinking beyond the ABC formulation and experimenting alternative discovery models in the discipline. Despite the simplicity and power of the ABC model, it also suffers from several limitations such as the sheer number of intermediate terms that exponentially expands the search space and producing a large number of target terms that are hard to interpret manually (*Smalheiser, 2012*). Even though the research in LBD have suggested various ways to overcome the aforementioned two limitations, most of these studies rely on similarity based measures to rank the target terms. This will result in LBD systems that merely detect *incremental* discoveries. In addition, the field requires to explore various *interestingness measures* that allows to customise the LBD output that cater different types of scientific investigations (*Smalheiser, 2012*).

With respect to other LBD discovery models that are enhanced based on ABC discovery structure include *AnC* model where $n = (B1, \ldots, Bn)$ (*Wilkowski et al., 2011*), combination of both *open* and *closed* discovery models (*Petrič et al., 2009*), *context-based ABC* model (*Kim & Song, 2019*), and *context-assignment-based ABC* model (*Kim & Song, 2019*). Moreover, recent studies have attempted to further explore alternative discovery models that deviate from the typical ABC discovery setting. These new directions include storytelling methodologies (*Sebastian, Siew & Orimaye, 2017b*), analogy mining (*Mower et al., 2016*), outlier detection (*Gubiani et al., 2017*), gaps characterisation (*Peng, Bonifield & Smalheiser, 2017*), and negative consensus analysis (*Smalheiser & Gomes, 2015*). For a comprehensive discussion of contemporary discovery models and future directions, please refer (*Smalheiser, 2017*; *Smalheiser, 2012*).

## Purpose of the review

Even though there are several review papers (*Gopalakrishnan et al., 2019*; *Henry & McInnes, 2017*; *Sebastian, Siew & Orimaye, 2017a*; *Ahmed, 2016*) published on LBD, the field still lacks systematic literature reviews. Therefore, the existing reviews merely cover a subset of LBD literature and do not provide a comprehensive classification of the LBD discipline. To address this gap, we present a large-scale systematic review by analysing 176 papers that were selected by manually analysing 475 papers. On the contrary to the existing traditional reviews, systematic reviews follow a rigorous and transparent approach to ensure the future replicability of results through the use of a clear systematic review protocol, and to minimise the bias in results by focusing on empirical evidence to present results, not preconceived knowledge (*Mallett et al., 2012*).

Another persistence research deficiency of other literature reviews is due to their limited and ad-hoc focus points. To date, none of the existing reviews focuses on the LBD workflow as a whole. Moreover, despite the importance of LBD components such as input, output, and evaluation, the existing reviews have not paid attention to critically analyse the state-of-the-art and the limitations of these components. To overcome these

two limitations, in this review we provide a sequential walk through of the entire LBD workflow by providing new insights into the LBD components such as input, output, and evaluation.

Furthermore, we have also observed that most of the existing reviews have restricted their scope only to medical-related LBD studies. Consequently, these reviews are lacking the discussions of LBD in non-medical and domain independent setting. To cater this issue, we have examined the LBD literature in both medical and non-medical domains in this review.

More specifically, our contributions are; (1) being the first systematic literature review that covers every component of the LBD workflow, (2) shedding light on components in LBD workflow such as input, output, and evaluation that have not been critically analysed or categorised by the existing reviews, (3) answer each of our research questions using novel, up-to-date and comprehensive categorisations compared to the existing reviews, and (4) critiquing LBD literature independently from domain without restricting to only medical-related LBD studies.

## METHODS

The overall process of this systematic review adheres the steps of *Systematic Literature Reviews in Computer Science* (*Weidt & Silva, 2016*) as illustrated in Fig. 1.

### Article retrieval process

We used six keywords and six databases to retrieve the articles for this review. Each keyword is searched in the title, abstract or keywords depending on the search options given by the databases. To ensure that we have not missed any useful articles, we also added the full reference list of a latest LBD review (*Henry & McInnes, 2017*). The article retrieval process with relevant statistics is summarised in Table 1.

### Article selection process

We only included journals and conference proceedings that are in the English language in our analysis. We excluded other types of articles such as reviews, books, book chapters, papers reporting lessons learned, keynotes, and editorials. We also eliminated the papers that provide the theoretical perspective of LBD as our research questions are focused to assess the LBD discipline in terms of computational techniques. We also excluded articles of page count 4 or below as such articles mainly contain research-in-progress. The entire article selection of this review was performed in three stages (*Weidt & Silva, 2016*); *Stage 1:* analysing only title and abstract, *Stage 2:* analysing introduction and conclusion, and *Stage 3:* read complete article and quality checklist. In total, we obtained 176 papers for this review (listed in https://tinyurl.com/selected-LBD-articles).

## REVIEW OVERVIEW

In this review we seek answers for eight research questions that are grouped into four categories by considering the workflow of LBD process as illustrated in Fig. 2.

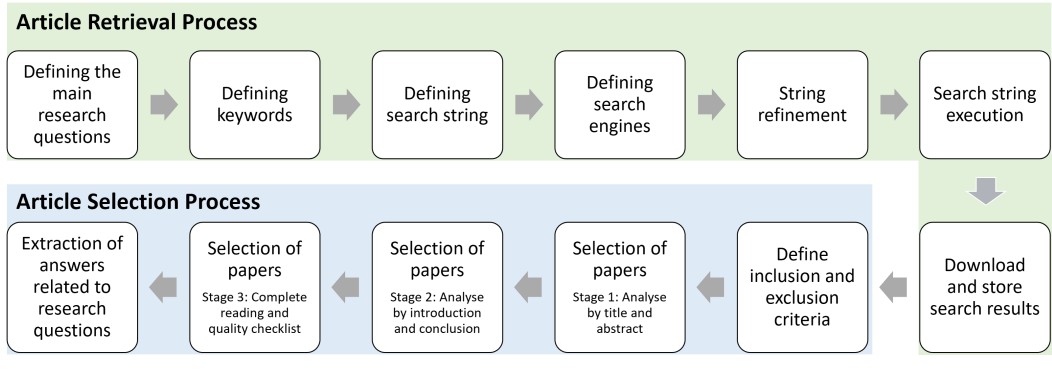

**Figure 1  Process of the systematic literature review.**

**Table 1  Statistics of the article retrieval process.**

| Keyword | Web of Science | Scopus | PubMed | ACM Digital Library | IEEE Xplore | Springer -Link | Total count |
|---|---|---|---|---|---|---|---|
| *Query 1*[a] | 161 | 68 | 75 | 15 | 15 | 8 | 342 |
| *Query 2*[b] | 14 | 0 | 4 | 1 | 2 | 1 | 22 |
| *Query 3*[c] | 14 | 0 | 0 | 0 | 0 | 1 | 15 |
| References from *Henry & McInnes (2017)* | | | | | | | 96 |
| **Total Article count** | | | | | | | **475** |

Notes.
[a] "literature based discovery" OR "literature based discoveries"
[b] "literature based knowledge discovery" OR "literature based knowledge discoveries".
[c] "literature related discovery" OR "literature related discoveries".

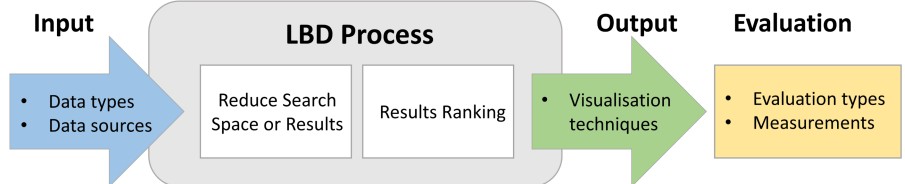

**Figure 2  Main components of the LBD workflow. .**

1. **Input Component** What are the data types considered for knowledge discovery? What are the data sources used in LBD research?

2. **Process component**

   What are the filtering techniques used in the LBD process?
   What are the ranking/thresholding mechanisms used in LBD literature?
   What are the domain independent and domain dependent resources utilised in LBD research?

3. **Output component**

   What are the visualisation techniques used to display the results in LBD research?

4. **Evaluation component**

What are the LBD evaluation types and their domain dependencies?

What are the quantitative measurements used to assess the effectiveness of the results?

To increase the readability of our review, we have cited a limited number of literature for each research question. However, a complete list of references that supports the proposed categorisations and conclusions of the research questions are listed in https://tinyurl.com/full-references.

## INPUT COMPONENT

This section analyses the input component of the LBD workflow to get an overview of the data structures and databases used in the literature.

### What are the data types considered for knowledge discovery?

LBD literature makes use of different *data types* as their input of the knowledge discovery process. The selection of the most suitable data type is one of the key design decisions, as they should represent the most important entities and relationships of the article to perform an efficient knowledge discovery. The *data types* used in the LBD literature can be categorised as follows.

*Title only:* Some LBD studies (*Swanson & Smalheiser, 1997*; *Cherdioui & Boubekeur, 2013*) have only considered the *article title* as the input of the knowledge discovery process. This input type selection might have influenced from Swanson's initial work as he only utilised the titles to uncover the hidden associations (e.g., *Raynaud* ↔*Fish Oil*). Even though the article title contains limited information, Sebastian et al. *Sebastian, Siew & Orimaye (2017b)* have reported that using only titles for knowledge discovery tend to produce better results compared to analysing abstracts.

*Title and Abstract:* The most common data type selection in literature is using both *title and abstract* (*Lever et al., 2017*; *Sebastian, Siew & Orimaye, 2017b*). The main reasons for this selection over full-text analysis could be; (1) *Reducing noise:* Typically the title and abstract include the most important concepts that best describe the study than considering the full-text, (2) *Data retrieval constraints:* Most APIs of the literature databases only support metadata retrieval, and (3) *Reducing computational complexity*: as the content of title and abstract is restricted the time and space complexities are reduced compared to full-text analysis.

*Full-text:* Few studies (*Lever et al., 2017*; *Vicente-Gomila, 2014*) have considered the entire content of articles as their input type. It has been reported that using full-text yields better results over title and abstract analysis (*Seki & Mostafa, 2009*). However, it is also important to pay attention as to what sections of the full-text need to be analysed to obtain better results. For instance, does analysing only the methodological-related sections of the article produce better results than analysing the entire article? Such sections-related analysis have not been preformed in LBD literature yet.

*Selected articles only:* While most of the studies have used data retrieved from literature database search engines (e.g., Medline) for analysis, Cameron et al. *Cameron et al. (2015)* have only considered the reference lists of Swanson's LBD publications. Considering only the 65 articles cited in Swanson's *Raynaud ↔ Fish Oil* LBD paper (*Swanson, 1986*) as the input of the knowledge discovery process can be taken as an example. However, since these reference lists are manually analysed and selected, whether this data type selection reflects the complexity of the real world data is doubtful.

*Entire literature database:* Several research studies (*Lever et al., 2017*; *Yang et al., 2017*) have considered the entire literature database as the LBD input without only limiting to articles retrieved for a given query (e.g., subset of the articles retrived for the query 'Fish oil'). Since the primary focus of LBD research is in the medical domain, the literature database that has been mainly considered for analysis is *Medline.* Additionally, other sources such as SemMedDB (*Cohen et al., 2014*) and PubMed Central Open Access Subset articles (*Lever et al., 2017*) have also been considered as the input.

*Keywords:* Some research approaches have employed the keywords of the articles as the input data type (*Pusala et al., 2017*; *Hu et al., 2010*). The mostly utilised keyword type is *Medical Subject Headings (MeSH)* that are associated with Medline records. It is considered that MeSH descriptors are accurate and medically relevant as National Library of Medicine (NLM) employs trained indexers to assign them to the Medline articles. Therefore, it is considered as a reliable source of representing the content of the article.

*Other metadata:* Few studies have analysed other metadata of the research articles such as author details (*Sebastian, Siew & Orimaye, 2017b*), publisher details (*Sebastian, Siew & Orimaye, 2015*) and reference details (*Kostoff et al., 2008a*) to glean additional clues for the possible links in the knowledge discovery process. The results prove that such metadata enhances the predictability of implicit knowledge associations (*Sebastian, Siew & Orimaye, 2017b*).

*Other traditional data types:* While the majority of the studies have focused only on the analysis of research papers, some approaches have been conducted using other traditional data types such as patents (*Vicente-Gomila, 2014*; *Maciel et al., 2011*), and TREC MedTrack collection of clinical patient records (*Symonds, Bruza & Sitbon, 2014*), case reports (*Smalheiser, Shao & Philip, 2015*) as their input to the LBD process.

*Non-traditional data types:* Few research studies have attempted to perform LBD using non-traditional data types such as tweets (*Bhattacharya & Srinivasan, 2012*), Food and Drug Administration (FDA) drug labels (*Bisgin et al., 2011*), Popular Medical Literature (PML) news articles (*Maclean & Seltzer, 2011*), web content (*Gordon, Lindsay & Fan, 2002*), crime incident reports (*Schroeder et al., 2007*) and commission reports (*Jha & Jin, 2016a*). Their results have proved the suitability of LBD discovery setting in a non-traditional context to elicit hidden links.

The *Data unit of analysis* denotes the types of data extracted from the above-discussed data types to represent the knowledge associations. Since most of the LBD research performed in medicine, the most common term representation is using *UMLS* and *MeSH* (*Lever et al., 2017*; *Preiss & Stevenson, 2017*). Apart from these two medical resources, other medical databases such as *Entrez Gene* (*Kim et al., 2016*), *HUGO* (*Petric et al., 2014*),

*LocusLink* (*Hristovski et al., 2005*), *OMIM* (*Hristovski et al., 2003*) and *PharmGKB* (*Kim & Park, 2016*) have also being used extract data units. LBD studies in other domains mainly consider *word or word phrases (n-grams)* as their term representation (*Qi & Ohsawa, 2016*) that have been extracted using techniques such as Part-Of-Speech (POS) tag patterns.

### What are the data sources used in LBD research?

*Medline/PubMed* is extensively being used as the main data source of the LBD literature (*Lever et al., 2017*). Additionally, other data sources such as *PubMed Central (PMC) Open Access* (*Ding et al., 2013*), *Science Direct* (*Vicente-Gomila, 2014*), *Web of Science* (*Sebastian, Siew & Orimaye, 2015*), *IEEE Xplore Digital Library* (*Qi & Ohsawa, 2016*), *Engineering Village* (*Kibwami & Tutesigensi, 2014*), *ProQuest* (*Kibwami & Tutesigensi, 2014*), *EBSCO Host* (*Kibwami & Tutesigensi, 2014*), *INSPEC* (*Ye, Leng & Guo, 2010*) have also been employed by several other LBD approaches to retrieve the articles for analysis.

The patent-based LBD studies (*Vicente-Gomila, 2014*), have considered patent databases such as *Thomson Innovation*, *United State Patent and Trade Mark Office (USPTO)* and *MAtrixware REsearch Collection (MAREC) patent document collection* to retrieve the data. Other conventional data sources include *clinical datasets* (*Dong et al., 2014*), *Gene Expression Omnibus (GEO) database* (*Hristovski et al., 2010*), *ArrayExpress (AE) database* (*Maver et al., 2013*), *Manually Annotated Target and Drug Online Resource (MATADOR)* (*Crichton et al., 2018*), *Biological General Repository for Interaction Datasets (BioGRID)* (*Crichton et al., 2018*), *PubTator* (*Crichton et al., 2018*), *Online Mendelian Inheritance in Man (OMIM)* (*Cohen et al., 2010*) and *TREC* (*Symonds, Bruza & Sitbon, 2014*).

Few non-English data sources such as *Chinese Social Sciences Citation Index* (*Su & Zhou, 2009*), *China Biology Medicine disks* (*Qian, Hong & An, 2012*), *Chinese Medicine Library and Information System* (*Yao et al., 2008*), *Traditional Chinese Medicine Database* (*Gao et al., 2016*) and *Chinese Journal Full-text database* (*Yao et al., 2008*) have also been utilised in LBD workflow.

The studies that have attempted to perform LBD in a non-traditional setting have extracted data from a variety of sources such as *Twitter* (*Bhattacharya & Srinivasan, 2012*), *DailyMed: FDA drug labels* (*Bisgin et al., 2011*), *Google news* (*Maclean & Seltzer, 2011*), and *World Wide Web (WWW)* (*Gordon, Lindsay & Fan, 2002*).

## PROCESS COMPONENT

This section outlines the two major elements of the *process* component; *filtering techniques* and *ranking/thresholding techniques*. Moreover, this section also discusses about the *resources* utilised in LBD workflow.

### What are the filtering techniques used in the LBD process?

It is vital to provide a *concise output* to the user that is easily interpretable by only including the most promising knowledge associations. To achieve this, the search space of the knowledge discovery should be reduced by eliminating spurious, general, uninteresting, or

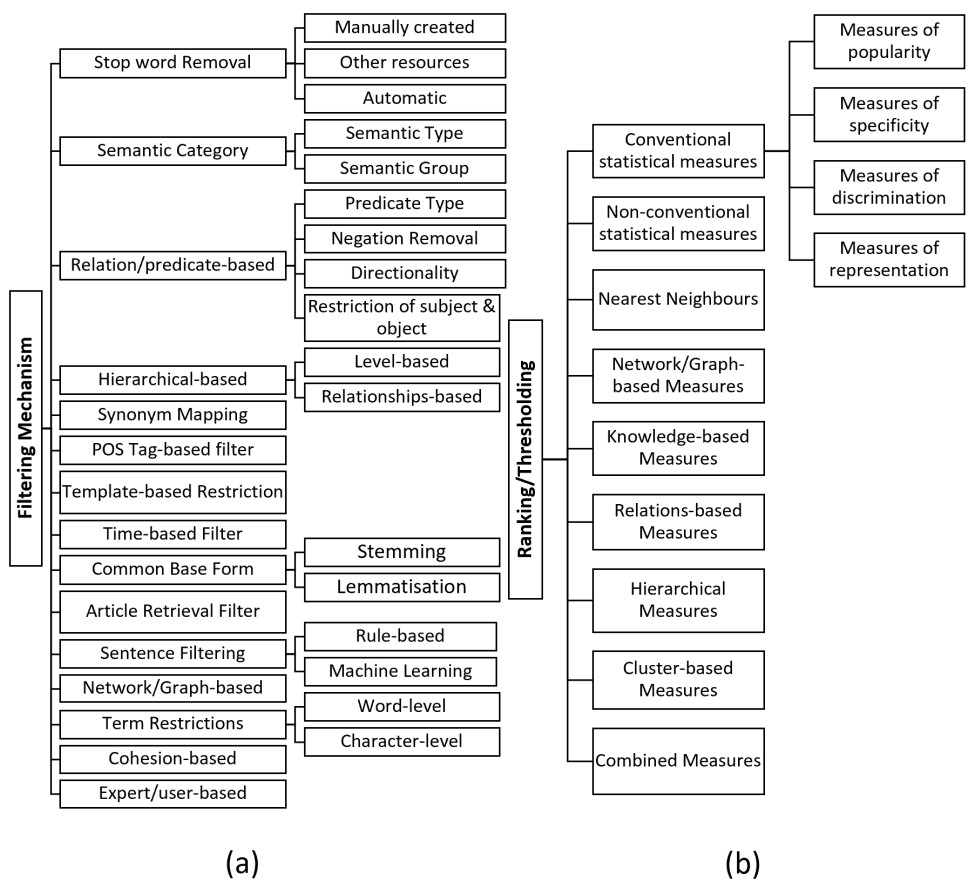

**Figure 3** **(A) Filtering techniques, (B) ranking/thresholding techniques.**

invalid terms/concepts. Different filtering techniques used in the literature are summarised in Fig. 3A.

*Stop word Removal:* Stop words typically denote non-topic general English terms. However, it could also include general terms used in the domain. For example, terms such as *"drug"*, *"treatment"* can be considered as general terms in the medical domain. Using stop words to remove uninformative terms is a popular filtering technique used (*Lever et al., 2017*; *Preiss & Stevenson, 2017*; *Sebastian, Siew & Orimaye, 2017b*). Stop word list could be either *manually created*, *obtained from other resources*, or *automatically generated*. (1) *Manually created:* A popular example of this category is the stop word list created for the *Arrowsmith* project (*Smalheiser, 2005*) that have nearly 9500 terms by 2006 (*Preiss & Stevenson, 2017*). However, manual development of a stop words is costly, and time-consuming. Moreover, since these stop word lists are highly domain dependent, their applicability is also limited. (2) *Obtained from other resources:* Other resources used to obtain stopword list include *NLTK toolkit* (*Lever et al., 2017*) and *Corpus of Contemporary American English* (*Lever et al., 2017*), and Nvivo (*Kibwami & Tutesigensi, 2014*). (3) *Automatically generated:* Some studies (*Preiss & Stevenson, 2017*; *Hu et al., 2010*) have automatically created their stop word lists by employing different techniques. The most common way is eliminating terms

that appear above a user-defined threshold (*Pratt & Yetisgen-Yildiz, 2003*). Different to the threshold-based removal, *Xun et al. (2017)* have followed *Law of conformity* to remove general terms by analysing the temporal change of terms, and *Jha & Jin (2016b)* have considered outliers of the box-plot as the general terms removal mechanism.

*Semantic Category Filter:* This technique typically utilises the *semantic type or group* information provided by *UMLS* (*Lever et al., 2017*; *Vlietstra et al., 2017*). UMLS currently provides 127 semantic types (https://semanticnetwork.nlm.nih.gov/SemanticNetworkArchive.html) and each medical concept is classified to one or more of these semantic types based on the relevance. Each semantic type is further classified into one or more of 15 UMLS semantic groups (https://semanticnetwork.nlm.nih.gov/download/SemGroups.txt). For example, *panic disorder* belongs to the semantic type *Mental or Behavioural Dysfunction* and *migraine* belongs to the semantic type *Disease and Syndrome*. Both these semantic types belong to the semantic group of *Disorders* (*Yetisgen-Yildiz & Pratt, 2009*). This filtering technique involves imposing selected semantic type or group to restrict the linking and target concepts of the knowledge discovery process. However, selecting the most suitable semantic type or group is very challenging as it varies according to the problem. If a too granular semantic category is selected, it may also remove the valid associations, and if a too broader semantic category is picked, it may not filter out all the meaningless associations.

*Relation/predicate Type Filter:* This filtering technique mostly consider the predications assigned using *SemRep* (*Cameron et al., 2015*; *Rastegar-Mojarad et al., 2015*). The typical procedure is to restrict the search space by eliminating uninteresting predicate types. For example, *Cohen et al. (2010)* have removed "PROCESS_OF" predication in their LBD process as it is less informative. Other types of predicate filtering techniques are; (1) removal of negated relations (*Rastegar-Mojarad et al., 2016*), (2) considering the directionality of the predicate (*Baek et al., 2017*), and (3) restricting the semantic type or group of the subject and object of the predication (i.e., subject-relation-object triplet) (*Hristovski et al., 2010*).

*Hierarchical Filter:* This technique utilises the hierarchical information such as *levels* and *relationships* of terms to filter out uninformative associations (*Shang et al., 2014*). The *levels* of *UMLS/MeSH hierarchy* are typically examined to remove broader terms. For example, Qian et al. *Qian, Hong & An (2012)* have eliminated terms in the first and second level of *MeSH tree* to remove less useful, broad associations. Another approach is to analyse the *hierarchical relationships* of the concepts to eliminate terms that are too close to the starting term. For instance, Pratt and Yetisgen-Yildiz *Yetisgen-Yildiz & Pratt (2006)* have eliminated terms in the *UMLS hierarchy* such as children, siblings, parents and grandparents as they have observed that these terms are closely related to the starting term, thus, do not form any interesting association.

*Synonym Mapping:* Mapping synonyms by grouping exactly or nearly equal terms of a given term is another technique used to reduce the results (*Lever et al., 2017*; *Baek et al., 2017*). To facilitate this, resources such as *UMLS* (*Preiss, Stevenson & Gaizauskas, 2015*), *MeSH* (*Van der Eijk et al., 2004*), *Entrez gene database* (*Liang, Wang & Wang, 2013*), *HUGO* (*Özgür et al., 2011*) have been utilised.

*POS Tag-based Filter:* Several studies have utilised POS tags to restrict the search space by limiting the terms to nouns (*Qi & Ohsawa, 2016*), nominal phrases (*Ittipanuvat et al., 2014*) or verbs (*Kim et al., 2016*). For example, Qi and Ohsawa *Qi & Ohsawa (2016)* have only extracted *nouns* as unigrams.

*Template-based Restriction:* Some studies (*Maver et al., 2013*; *Cohen et al., 2012*) have reduced the search space by only extracting the associations that adhere to the imposed *rules/templates*. For example, two forms of *discovery patterns* were defined by *Hristovski et al. (2006)* to restrict the detected associations that are in accordance with the templates of the two defined patterns.

*Time-based Filter:* *Smalheiser (2005)* have considered the time factor of the associations to reduce the search space of results. More specifically, given a user-defined year, only the associations that appear first time after the year (or before) have been considered as a filter. In addition, monitoring the temporal behaviours of words (*Xun et al., 2017*) have also been used to remove unnecessary terms.

*Common Base Form:* Deriving a common base form of the term to reduce the vocabulary space is another technique used in the literature. To facilitate this, the two popular techniques; *stemming* (*Sebastian, Siew & Orimaye, 2015*) and *lemmatisation* (*Song, Heo & Ding, 2015*) have been used in the literature.

*Article Retrieval Filter:* Several studies (*Cherdioui & Boubekeur, 2013*; *Ittipanuvat et al., 2014*) have attempted to limit the number of articles that need to be analysed through the LBD process to reduce the search space. For instance, *Petrič et al. (2012)* have only considered the *outlier* documents for analysis without analysing all the documents derived from the search query.

*Sentence Filter:* Some studies (*Hossain et al., 2012*; *Özgür et al., 2010*) have only picked specific sentences from the text to analyse. For example, *Özgür et al. (2010)* have only picked sentences from abstracts that describe gene interactions for the analysis. For a sentence to qualify as a potential interaction sentence, the authors have followed a rule-based mechanism. Moreover, *Hossain et al. (2012)* have employed machine learning to select sentences by training a Naïve Bayes classifier to differentiate *context* and *results* sentences in abstracts.

*Network-based Filter:* The network-based LBD approaches have utilised different techniques to reduce the size of the network. For example, *Cairelli et al. (2015)* have filtered their network by setting degree centrality and edge-occurrence frequency thresholds. Furthermore, *Kastrin, Rindflesch & Hristovski (2014)* have performed Pearson's Chi-Square test to detect if a particular connection occurs more often by chance. *Ittipanuvat et al. (2014)* have removed nodes that are not connected with any node in Largest Connected Components (LCC) of the graph.

*Term Restrictions:* Some studies have restricted terms in *word-level* and *character-level* to reduce the vocabulary space. Removal of unigrams from the analysis can be considered as an example for word-level restriction (*Thaicharoen et al., 2009*; *Gordon, Lindsay & Fan, 2002*). LBD studies (*Roy et al., 2017*; *Kibwami & Tutesigensi, 2014*) that have removed terms less than three characters in their LBD process can be considered as an example for

character-level restrictions. However, since this filter does not consider semantic aspects of the terms into consideration, valuable short terms will be removed from the vocabulary.

*Cohesion-based filter:* Given two linking terms that are most similar, *Smalheiser (2005)* hypothesises that the term with a more narrow focus is the most useful. Hence, this filter calculates a *cohesion score* to select most granular-level terms as the results.

*Expert/user-based filtering:* Expert/user-based filtering (*Gubiani et al., 2017*; *Preiss & Stevenson, 2017*) involves the decision of an expert/user to remove uninteresting associations. For example, most of the *semantic category filter* requires user-defined semantic types/groups to perform the filtering. As described in 'Semantic Category Filter' technique, this selection is crucial as a more restrictive semantic category selection would risk at losing valid and informative associations whereas less restrictive semantic category selection would result in a noisy output. As a result, the success of these approaches greatly depends on the experience and prior knowledge of the user.

## What are the ranking/thresholding mechanisms used in LBD literature?

Term ranking/thresholding is an important component of the LBD process as it should downweigh or remove noisy associations and upweight or retain the interesting and significant knowledge associations when ordering the terms. More specifically, these measures are used in two ways. (1) *Thresholding:* prune away uninteresting associations during the filtering process (e.g., setting a threshold to remove general terms), (2) *Ranking:* rank the selected set of associations based on their significance (e.g., rank the most frequent terms in the top of the list). Outlined below are the ranking schemes used in the discipline (See Fig. 3B).

Considering *conventional statistical measures* to rank/threshold terms is common in literature. These measures can be broadly divided into four categories (*Aizawa, 2003*) based on how they are mathematically defined; (1) *Measures of popularity:* these measures denote the frequencies of terms or probability of occurrences (e.g., concept frequency), (2) *Measures of specificity:* this category denotes the entropy or the amount of information of terms (e.g., mutual information), (3) *Measures of discrimination:* how terms are contributing to the performance of a given discrimination function is represented through these measures (e.g., information gain), and (4) *Measures of representation:* these measures denote the usefulness of terms in representing the document that they appear (e.g., TF-IDF).

Examples for conventional statistical measures used in LBD studies are; *Token frequency* (*Gordon & Lindsay, 1996*), *Average token frequency* (*Ittipanuvat et al., 2014*), *Relative token frequency* (*Lindsay, 1999*), *Document/record frequency* (*Gordon & Lindsay, 1996*), *Average Document Frequency* (*Ittipanuvat et al., 2014*), *Relative Document Frequency* (*Thaicharoen et al., 2009*), *TF-IDF* (*Maciel et al., 2011*), *Mutual Information* (*Loglisci & Ceci, 2011*), *z-score* (*Yetisgen-Yildiz & Pratt, 2006*), *Information Flow* (*Bruza et al., 2006*), *Information Gain* (*Pusala et al., 2017*), *Odds Ratio* (*Bruza et al., 2006*), *Log Likelihood* (*Bruza et al., 2006*), *Support* (*Hristovski et al., 2005*), *Confidence* (*Hristovski et al., 2003*), *F-value of support and confidence* (*Hu et al., 2010*), *Chi-Square* (*Jha & Jin, 2016b*), *Kulczynski* (*Jha & Jin, 2016a*), *Cosine* (*Baek et al., 2017*), *Equivalence Index* (*Stegmann & Grohmann, 2003*),

*Coherence* (*Pusala et al., 2017*), *Conviction* (*Pusala et al., 2017*), *Klosgen* (*Pusala et al., 2017*), *Least Contradiction* (*Pusala et al., 2017*), *Linear-Correlation* (*Pusala et al., 2017*), *Loevinger* (*Pusala et al., 2017*), *Odd Multiplier* (*Pusala et al., 2017*), *Piatetsky-Shapiro* (*Pusala et al., 2017*), *Sebag-Schoenauer* (*Pusala et al., 2017*), *Zhang* (*Pusala et al., 2017*), *Jaccard Index* (*Yang et al., 2017*), *Dice Coefficient* (*Yang et al., 2017*), and *Conditional Probability* (*Seki & Mostafa, 2009*).

Additionally, *non-conventional statistical measures* such as such as *Average Minimum Weight (AMW)* (*Yetisgen-Yildiz & Pratt, 2009*), *Linking Term Count with AMW (LTC-AMW)* (*Yetisgen-Yildiz & Pratt, 2009*), *Averaged Mutual Information Measure (AMIM)* (*Wren, 2004*), *Minimum Mutual Information Measure (MMIM)* (*Wren, 2004*) have also been proposed in discipline to rank the potential associations. In comparison with *AMW* and *Literature Cohesiveness*, *Yetisgen-Yildiz & Pratt (2009)* have reported that they gained improved performance with *LTC-AMW* measure (*Swanson, Smalheiser & Torvik, 2006*). Other types of ranking and thresholding categories used in LBD literature are summarised below.

*Nearest Neighbours:* In this category, the score of the association is decided by analysing its nearest neighbours. Such analysis is typically performed in distributional semantic models by employing measures such as *Cosine* (*Gopalakrishnan et al., 2017*), *Euclidian distance* (*Van der Eijk et al., 2004*), and *information flow* (*Bruza et al., 2006*).

*Network/Graph-based Measures:* Network/graph-based measures analyse node and edge-level attributes to score the associations. Examples of measures that represent this category include *Degree centrality* (*Goodwin, Cohen & Rindflesch, 2012*), *Eigenvector centrality* (*Özgür et al., 2010*), *Closeness centrality* (*Özgür et al., 2011*), *Betweenness centrality* (*Özgür et al., 2010*), *Common Neighbours* (*Kastrin, Rindflesch & Hristovski, 2014*), *Jaccard Index* (*Kastrin, Rindflesch & Hristovski, 2014*), *Preferential Attachment* (*Kastrin, Rindflesch & Hristovski, 2014*), *Personalised PageRank* (*Petric et al., 2014*), *Personalised Diffusion Ranking* (*Petric et al., 2014*), and *Spreading Activation* (*Goodwin, Cohen & Rindflesch, 2012*).

*Knowledge-based Measures:* This category denotes the scoring measures such as *MeSH-based Literature cohesiveness* (*Swanson, Smalheiser & Torvik, 2006*), *semantic type co-occurrence* (*Jha & Jin, 2016b*), *chemDB atomic count* (*Ijaz, Song & Lee, 2010*), and *chemDB XLogP* (*Ijaz, Song & Lee, 2010*) that involve the knowledge from structured resources to rank the associations. The advantage of these measures is that they entangle semantic aspects into consideration to decide the potentiality of the association.

*Relations-based Measures:* Relations/predicate based measures (a sub-class of *knowledge-based measures*) analyse the relations extracted from resources such as SemRep to rank/threshold associations. Scoring measures such as *Semantic relations frequency* (*Hristovski et al., 2010*), *Predicate independence* (*Rastegar-Mojarad et al., 2015*), *Predicate interdependence* (*Rastegar-Mojarad et al., 2015*), *Edge frequency-based weight* (*Kim et al., 2016*), *Edge traversal probability* (*Vlietstra et al., 2017*), *Relationship traversal probability* (*Vlietstra et al., 2017*), *Source traversal probability* (*Jha & Jin, 2016b*), and *Impact Factor* (*Huang et al., 2016*) are examples of this category.

*Hierarchical Measures:* This category is another sub-class of *knowledge-based measures* that utilise hierarchical information of taxonomies such as *UMLS*, and *MeSH* to derive

the rankings. _Child-to-parent and parent-to-child predications_ (_Seki & Mostafa, 2009_), and _MeSH tree code depth_ (_Gopalakrishnan et al., 2017_) can be considered as examples.

_Cluster-based Measures:_ In this category, cluster similarities are measured using techniques such as _Intra-cluster similarity_ (_Cameron et al., 2015_), _Jaccard Index_ (_Ittipanuvat et al., 2014_), _Inclusion Index_ (_Ittipanuvat et al., 2014_), _Dice coefficient_ (_Ittipanuvat et al., 2014_), _Cosine_ (_Ittipanuvat et al., 2014_), _Cosine similarity of tf-idf_ (_Ittipanuvat et al., 2014_), and _Cosine similarity of tf-lidf_ (_Ittipanuvat et al., 2014_) to derive the ranking scores of associations.

_Combined Measures:_ The idea of combined measures is to integrate multiple characteristics of an association to decide its potential ranking. For example, _Torvik & Smalheiser (2007)_ have utilised machine learning techniques to combine seven characteristics of an association such as _absolute and relative term frequencies_, _cohesion_, _recency_, etc to obtain the final ranking score. _Song, Heo & Ding (2015)_ have also proposed a combined ranking measure by considering an average of three _semantic similarity measures_, and _SemRep score_. The characteristics that have been considered in the study of _Ijaz, Song & Lee (2010)_ include _UMLS semantic type_, _structural similarity_, _chemDB atomic count_, and _chemDB XLogP_. Similarly, _Gopalakrishnan et al. (2017)_ have also introduced a combined ranking measure by integrating global (_node centrality_ and _MeSH tree code depth_) and local (_semantic co-occurrence_ and _betweenness centrality_) measures. Overall, combined ranking measure are more flexible as they rely on multiple characteristics to prioritise the derived associations.

## What are the domain independent and domain dependent resources utilised in LBD research?
### Domain dependent resources

Since the majority of LBD research are in Medicine, we refer _medical resources_ as domain dependent resources. These resources are further categorised as; (1) _Resources that provide background domain knowledge_, and (2) _Resources that are used in content analysis_.

_Resources to acquire background domain knowledge:_ The main purposes of extracting the domain knowledge are; (1) _input data preparation_ (e.g., concept extraction), (2) _filtering the noisy, uninteresting or unrelated associations_ (e.g., semantic type filtering), (3) _prepare a ranking mechanism_ (e.g., hierarchical ranking), (4) _evaluate the results_ (e.g., compare results with curated databases), and (5) _training data preparation_. The popular domain dependent resources used in the discipline are;

- UMLS: (_Lever et al., 2017_; _Vlietstra et al., 2017_; _Preiss & Stevenson, 2017_)
- MeSH: (_Baek et al., 2017_; _Xun et al., 2017_; _Pusala et al., 2017_)
- SemMedDB/Semantic Medline: (_Vlietstra et al., 2017_; _Cairelli et al., 2015_)
- Gene Ontology: (_Baek et al., 2017_; _Huang et al., 2016_; _Kim et al., 2016_)
- Entrez Gene Database: (_Baek et al., 2017_; _Liang, Wang & Wang, 2013_; _Kwofie et al., 2011_)
- Kyoto Encyclopedia of Genes and Genomes (KEGG): (_Kwofie et al., 2011_)
- HGNC/HUGO: (_Petric et al., 2014_; _Ding et al., 2013_; _Maciel et al., 2011_)
- UNIPROT: (_Baek et al., 2017_; _Vlietstra et al., 2017_), Swiss-Prot (_Jelier et al., 2008_)

- Therapeutic Target Database (TTD): (*Yang et al., 2017*; *Maciel et al., 2011*)
- LocusLink: (*Smalheiser, 2005*; *Hristovski et al., 2003*)
- Online Mendelian Inheritance in Man (OMIM) (*Hristovski et al., 2003*; *Wren et al., 2004*)
- Drug Bank: (*Vlietstra et al., 2017*; *Maciel et al., 2011*; *Ding et al., 2013*)
- Comparative Toxicogenomics Database (CTD): (*Vlietstra et al., 2017*; *Yang et al., 2017*)
- BioGRID: (*Huang et al., 2016*; *Crichton et al., 2018*)
- Gene2pubmed: (*Cheung, Ouellette & Wasserman, 2012*; *Roy et al., 2017*)
- Drugs.com: (*Maciel et al., 2011*; *Banerjee et al., 2014*)
- SIDER Side Effect Resource: (*Vlietstra et al., 2017*; *Shang et al., 2014*)

Additionally, other medical resources such as *Medical Dictionary for Regulatory Activities (MedDRA)* (*Bisgin et al., 2011*), *Reactome Pathway Database* (*Kwofie et al., 2011*), *Orphanet* (*Baek et al., 2017*), *Human Metabolome Database (HMDB)* (*Baek et al., 2017*), *Lipid Maps* (*Baek et al., 2017*), *MassBank* (*Baek et al., 2017*), *DailyMed* (*Vlietstra et al., 2017*), *miRBase* (*Huang et al., 2016*), *miRGate* (*Huang et al., 2016*), *Transcriptional Regulatory Relationships Unraveled by Sentence-based Text mining (TRRUST)* (*Huang et al., 2016*), *PAZAR* (*Huang et al., 2016*), *Biomedical Knowledge Repository (BKR)* (*Cameron et al., 2015*), *MEDI* (*Shang et al., 2014*), *Tanabe-Wilbur list* (*Smalheiser, 2005*), *ChemDB* (*Ijaz, Song & Lee, 2010*), *BioVerb* (*Kim et al., 2016*), *AIMED* (*Özgür et al., 2010*), *CB* (*Özgür et al., 2010*), *STRING* (*Petric et al., 2014*), *ToppGene* (*Petric et al., 2014*), *Endeavour* (*Petric et al., 2014*), *MIPS* (*Liang, Wang & Wang, 2013*), *Proteomics Standards Initiative Molecular Interactions (PSI-MI)* (*Song, Heo & Ding, 2015*), *Cell Line Knowledge Base (CLKB)* (*Song, Heo & Ding, 2015*), *Observational Medical Outcomes Partnership (OMOP)* (*Mower et al., 2016*), *METADOR* (*Crichton et al., 2018*), *Animal Transcription Factor Database (AnimalTFDB)* (*Roy et al., 2017*), *RxNorm* (*Malec et al., 2016*), *Vaccine Ontology (VO)* (*Özgür et al., 2011*), *Gene Reference Into Function (GeneRIF)* (*Cheung, Ouellette & Wasserman, 2012*), *Homologene* (*Jelier et al., 2008*), *Pharmacogenomics Knowledgebase (PharmGKB)* (*Kim & Park, 2016*), *Chinese Medical Terminology* (*Qian, Hong & An, 2012*), *Food and Drug Administration approved drug names* (*Wren, 2004*), *Rush University Medical Center's health encyclopedia* (*Banerjee et al., 2014*) have also been employed in LBD workflow.

Our analysis reveals that *UMLS* and *MeSH* are most extensively used as the domain dependent resources in the literature[3]. The databases such as *SemMedDB/Semantic Medline*, *Gene Ontology*, *Entrez Gene Database* and *HUGO/HGNC* are also popular among other resources.

*Resources for content analysis:* The following resources have been used in LBD systems to process and analyse the content.

- MetaMap (Medical concept extraction): (*Preiss & Stevenson, 2017*; *Preiss & Stevenson, 2016*; *Cairelli et al., 2015*)
- SemRep (Semantic predications extraction): (*Vlietstra et al., 2017*; *Preiss, Stevenson & Gaizauskas, 2015*)
- Genia Tagger (Biological NER): (*Lever et al., 2017*; *Özgür et al., 2010*)
- ABNER (Biological NER): (*Liang, Wang & Wang, 2013*)

- Peregrine software (Biological NER): (*Jelier et al., 2008*)
- DAVID tool (Gene annotation enrichment analysis): (*Maver et al., 2013*; *Özgür et al., 2010*)
- RankProd Package (Meta analysis): (*Maver et al., 2013*)
- BioTeKS Text Analysis Engine (Text annotation): (*Berardi et al., 2005*)
- PubTator (PubMed citations annotation): (*Crichton et al., 2018*)
- MedLEE (Structure and encode clinical reports): (*Malec et al., 2016*)
- BioMedLEE (Semantic predications extraction): (*Hristovski et al., 2006*)
- EpiphaNet (Interactive visual representation): (*Malec et al., 2016*)
- SciMiner (literature mining and functional enrichment analysis): (*Hur et al., 2010*)
- Biovista (Drug repurposing, Systems literature analysis environment): (*Persidis, Deftereos & Persidis, 2004*)

Among the content analysis tools, we observed that *MetaMap* and *SemRep* are the most popular selections[3]. *MetaMap* is a tool that recognises *UMLS* concepts in the text whereas *SemRep* is used to extract semantic predications from the text. The predications in SemRep are formal representations of text content that comprises of *subject–predicate–object* triplets.

### Domain independent resources

In this section, we summarise the resources that can be used in a cross domain LBD setting. For Named Entity Recognition (NER) resources such as *GATE* (*Loglisci & Ceci, 2011*), *PKDE4J* (*Baek et al., 2017*), *Open Calais* (*Jha & Jin, 2016a*), *Sementax* (*Jha & Jin, 2016a*) and *Lingpipe* (*Hossain et al., 2012*) have been employed in LBD literature.

Other text analytics resources include *NLTK*: to identify Noun Phrases (*Sebastian, Siew & Orimaye, 2017b*) and stop words (*Lever et al., 2017*), *ReVerb:* to extract relations (*Preiss, Stevenson & Gaizauskas, 2015*), *Stanford parser:* for dependency tree parsing (*Sang et al., 2015*) and extract relations (*Preiss, Stevenson & Gaizauskas, 2015*), *Stanford CoreNLP:* for sentence boundary detection, POS tagging and lemmatisation (*Song, Heo & Ding, 2015*), *WordNet:* for word sense disambiguation (*Sebastian, Siew & Orimaye, 2017b*), *RacerPro:* for logical and rule-based reasoning (*Guo & Kraines, 2009*), *Link Grammar Parser:* for sentence parsing (*Ijaz, Song & Lee, 2010*), *Vantage Point:* for document clustering, auto-correction mapping and factor matrix analysis (*Kostoff, 2011*), *Nvivo:* to extract terms, stop words, coding and matrix coding queries (*Kibwami & Tutesigensi, 2014*), *CLUTO:* for document clustering (*Kostoff et al., 2008b*), *Lucene:* for information retrieval (*Malec et al., 2016*), and *OntoGen:* for topic ontology construction (*Petrič et al., 2009*).

To facilitate tasks such as network construction and visualisation, the following resources have been utilised in the literature; *Neo4j* (*Vlietstra et al., 2017*), *JUNG* (*Kim et al., 2016*), *Gephi* (*Song, Heo & Ding, 2015*), *NetworkX* (*Wilkowski et al., 2011*), and *Large Graph Layout (LGL)* (*Ittipanuvat et al., 2014*).

The importance of using the aforementioned resources in the LBD system is that they support the system's functionality not only to medical domain, but also to a wide variety of other domains. To date, such domain independent LBD methodologies have been rarely experimented.

## OUTPUT COMPONENT

This section discusses the existing LBD output types, their drawbacks and the important characteristics that need to be fulfilled in terms of output visualisation to meet the objectives of the discipline.

### What are the visualisation techniques used to display the results in LBD research?

The most commonly used output of LBD systems is a ranked list of associations (*Gubiani et al., 2017*; *Baek et al., 2017*) where the top associations reflect the most probable knowledge links. However, providing merely a ranked list may not be the best way of visualising the results due to the following two reasons; (1) ranked associations are isolated in nature and do not provide an overall picture of all the suggested associations, and (2) ranked associations do not reflect how they are linked with the start and/or target concepts to better understand the association. As a result, the user needs to manually analyse the ranked associations individually to get an overview of the entire results and to interpret the linkage of the proposed associations with the start and/or target concepts. This points out the importance of exploring better visualisation techniques that can reduce the manual walkthroughs the user requires to perform. Discussed below are other visualisation techniques employed in the literature.

*Group based on semantic type:* In Manjal LBD system (*Srinivasan, 2004*), the outputted terms are organised by UMLS semantic type and ranked based on its estimated potential within the semantic type.

*Rank based on templates:* SemBT LBD system (*Hristovski et al., 2010*) ranks the identified novel associations using frequency of semantic relations (relation triplets) by specifying the subject and object of the relation. *Ijaz, Song & Lee (2010)* have ranked the detected associations based on an information model that includes substance, effects, processes, disease and body part.

*Graph-based visualisations:* Several studies have utilised graphs to visualise their LBD results. For instance, *Kim et al. (2016)* have used directed gene-gene network to clearly illustrate the discovery pathways suggested by their LBD methodology. A more advanced graph-based visualisation was proposed by *Cameron et al. (2015)* that outputs multiple context driven sub-graphs. Since the graph is divided into subgraphs by grouping the paths with similar context, the results can be easily interpreted by the user.

*Ranking the discovery pathways:* From LBD perspective, this technique can also be viewed as the output of *AnC* model. While *graph-based visualisations* display graphs as output, this technique only lists down the potential paths from the graph. Examples of this category include the study of *Wilkowski et al. (2011)* where the graph paths with high degree centrality are shown as the output, and the study of *Kim et al. (2016)* that considers the shortest paths in the graph as the output.

*Story chain construction:* *Hossain et al. (2012)* have attempted to build story chains by focusing on biological entities in PubMed abstracts. Their storytelling algorithm provides new insights to LBD visualisation and can be viewed as a next step of the *Ranking the discovery pathways* technique.

*Word clouds: Malec et al. (2016)* have used word clouds to present their results where the font size is proportionate to term frequencies.

*Matrix-like visualisation: Qi & Ohsawa (2016)* have proposed a matrix-like visualisation to detect mixed topics of their experiments. Moreover, they have also performed a user-based evaluation by providing their visualisation to the users to detect and interpret the mixed topics.

*Using Existing Tools:* Some studies have utilised existing tools such as Semantic Medline (*Miller et al., 2012*), OntoGen (*Petrič et al., 2012*), and EpiphaNet (*Cohen, Schvaneveldt & Rindflesch, 2009*), Biolab Experiment Assistant (BAE) (*Persidis, Deftereos & Persidis, 2004*) for LBD visualisation.

Improving output visualisation is an essential component of the LBD workflow as it highly influences the user acceptance of the system. However, the existing literature has a little contribution towards output visualisation. This suggests the importance of involving Human Computer Interaction (HCI) techniques in the field. Some important characteristics that should be taken into consideration when developing a visualisation technique are; (1) concise output, (2) easily interpretable, (3) less complex, (4) visually attractive, and (5) assist users to gain new insights. Moreover, it is also vital to evaluate the efficiency of the visualisation technique by performing user-based evaluations (*Santos, 2008*). For instance, organising sessions for the participants to use the LBD tool (*Cohen et al., 2010*), observing how they interact with the tool and obtaining their feedback. Santos *Santos (2008)* suggests two types of participants for such evaluations; *target users* and *graphic designers*. The author points out that the target users will assist to elicit new ideas whereas graphic designers will detect problems and provide suggestions with visual aspects. Furthermore, another interesting avenue is to involve target users with different level of expertise (i.e., expert vs. novice) to evaluate how users with each level of expertise interact and benefit with the LBD process (*Qi & Ohsawa, 2016*).

## EVALUATION COMPONENT

### What are the LBD evaluation types and their domain dependencies?

Evaluating the effectiveness of the LBD results is challenging and remains to be an open issue. The main reason for this is that the LBD process detects novel knowledge that has not been publicly published anywhere and thus needs to be proven that they are useful. Moreover, there are no comprehensive gold standard datasets or consistent formal evaluation approaches in LBD (*Ganiz, Pottenger & Janneck, 2005*). This review provides an in-depth classification of the existing evaluation techniques as summarised below.

### *Evidences-based evaluation*

This category of evaluation asserts if a given association is accurate by using evidence from reliable sources such as existing discoveries, literature or curated databases.

*Replicating existing medical discoveries:* By far, this is the most commonly used evaluation technique. It measures the capability of the LBD methodology to reproduce the popular historical discoveries (see Table 2). The most popular selections of discovery replication are Swanson's initial two medical discoveries; *Raynaud ↔Fish Oil* and *Migraine*

**Table 2    Replicated discoveries in the literature.**

| Replicated Discovery | Past Studies |
|---|---|
| Migraine ↔Magnesium | *Xun et al. (2017), Preiss & Stevenson (2017), Sebastian, Siew & Orimaye (2017b), Qi & Ohsawa (2016), Song, Heo & Ding (2015)* |
| Raynaud ↔Fish Oil | *Xun et al. (2017), Preiss & Stevenson (2017), Sebastian, Siew & Orimaye (2017b), Song, Heo & Ding (2015), Preiss, Stevenson & Gaizauskas (2015)* |
| Indomethacin ↔Alzheimer's | *Xun et al. (2017), Preiss & Stevenson (2017), Preiss, Stevenson & Gaizauskas (2015), Cameron et al. (2015), Sang et al. (2015)* |
| Schizophrenia ↔Calcium-Independent Phospholipase A2 | *Xun et al. (2017), Preiss & Stevenson (2017), Preiss, Stevenson & Gaizauskas (2015), Cameron et al. (2015), Srinivasan (2004)* |
| Alzheimer's ↔Estrogen | *Preiss & Stevenson (2017), Preiss, Stevenson & Gaizauskas (2015), Cameron et al. (2015), Preiss (2014)* |
| Magnesium deficiency ↔Neurologic | *Preiss & Stevenson (2017), Preiss, Stevenson & Gaizauskas (2015), Preiss (2014)* |
| Thalidomide ↔Chronic Hepatitis C | *Kwofie et al. (2011), Jelier et al. (2008)* |
| Testosterone ↔Sleep | *Cameron et al. (2015), Goodwin, Cohen & Rindflesch (2012)* |
| Somatomedin C ↔Arginine | *Swanson & Smalheiser (1997), Preiss (2014)* |
| Chlorpromazine ↔Cardiac Hypertrophy | *Cameron et al. (2015)* |
| Diethylhexyl (DEHP) ↔Sepsis | *Cameron et al. (2015)* |
| Sleep ↔Depression | *Goodwin, Cohen & Rindflesch (2012)* |

↔*Magnesium*[3]. The normal procedure used for discovery replication is to only use the literature before the original paper of discovery as the input data of the LBD process and to verify if the mentioned associations detected in the original paper could be replicated. For example, if we consider Swanson's *Raynaud ↔Fish Oil* to replicate, the literature prior to 1986 (the published year of the paper) should only be considered.

However, discovery replication may not be the most effective way of evaluating the LBD methodology due to the following reasons. (1) These existing discoveries have not developed rigorously as a gold standard (*Ganiz, Pottenger & Janneck, 2005*). For example, in Swanson's *Raynaud ↔Fish Oil* discovery, he merely suggests three novel intermediate connections. No evidence suggest that these connections identified through his trial and error approach can be seen as the only existing novel associations that connect the two domains, (2) Only focusing on one particular discovery might result in a system that performs well for that problem, but not for other problems even in the same domain (overfitting) (*Yetisgen-Yildiz & Pratt, 2009*). For example, Swanson and Smalheiser *Swanson & Smalheiser (1997)* have replicated medical discoveries to evaluate *Arrowsmith* LBD system. The overfitting of their model is proved by the failure of it in recognising the links of *Somatomedin-C ↔Arginine* (*Swanson, 1990*). As a result, it is important to accompany other evaluation techniques along with *discovery replication* to measure the true efficiency of the proposed methodology.

*Time-sliced evaluation:* Time-sliced method evaluates the ability of LBD methodology to predict future co-occurrences based on a time-sliced dataset (*Lever et al., 2017; Yang et*

*al., 2017*). Currently this is the most objective evaluation technique in the discipline that attempts to alleviate the following key issues (*Yetisgen-Yildiz & Pratt, 2009*).

(1) Discovery replication is limited to the associations defined in that particular discovery and merely evaluates the ability of the methodology to recreate these specific associations. As a result, the remaining associations in the LBD output are not assessed. This makes it difficult to estimate the overall performance of the LBD system. Instead, time-sliced evaluation evaluates the complete list of associations outputted from the LBD system. (2) Most of LBD systems consider one or two existing medical discoveries to replicate. Hence, the true generalisability of their methodologies is not reflected. To overcome this issue, time-sliced evaluation is designed in a way it is repeatable for many starting concepts without only limiting to one or two existing medical discoveries. For example, *Yetisgen-Yildiz & Pratt (2009)* have considered 100 starting concepts for evaluation. (3) When replicating existing medical discoveries, the required intermediate and target terms are known in advance. As a result, the parameters of the system can be tuned in a way to obtain these terms which result in a system that performs well only for that discovery, but not in other cases. However, time-sliced evaluation is independent of prior knowledge as it does not require to know the output in advance which assists to perform an unbiased evaluation. (4) When replicating medical discoveries or in expert-based evaluation, it is difficult to compare the performance of different LBD systems. For example, if two systems claim that they could successfully replicate a particular discovery, it is hard to determine the most efficient system. Similarly, when incorporating expert decisions for evaluation, it is hard to quantify the results and compare against other LBD systems. As a result, time-sliced evaluation provides a platform to quantitatively compare the LBD outcomes with other systems.

This technique requires a *cut-off-date* to divide the dataset into two segments named as *pre-cut-off* (data before the specified cut-off date) and *post-cut-off* (data after the cut-off date). The pre-cut-off segment is treated as the training set, where the LBD system is employed to output the potential novel associations. Afterwards, the post-cut-off segment is utilised to develop the gold standard dataset to evaluate the produced associations. The gold standard dataset is created by identifying associations present in the post-cut-off set and absent in the pre-cut-off set. More specifically, it verifies whether the identified potential associations from the LBD process have taken place in the future. Therefore, the selection of the cut-off-date is crucial because it decides the time period that turns a hypothesis into a true discovery (*Yetisgen-Yildiz & Pratt, 2009*).

*Manual literature search:*

Some studies have verified whether the produced associations are meaningful by manually searching the research articles that provide evidences of the existence of the specified association (*Yang et al., 2017*; *Xun et al., 2017*).

*Intersection evaluation:*

This approach checks if the identified association has been co-occurred with the initial concept in any of the literature databases (e.g., Web of Science) or other sources (e.g., UseNet), and remove already known associations to filter out the novel associations

(*Gordon, Lindsay & Fan, 2002*; *Bhattacharya & Srinivasan, 2012*). Afterwards, filtered novel associations are qualitatively evaluated.

*Derive reference sets from literature:* In this technique, the methodology is evaluated by using reference sets created using past literature. For example, in the study of *Vlietstra et al. (2017)*, they have developed the reference set from the results of a systematic literature review to compare their results. In the work of *Malec et al. (2016)*, they have utilised curated drug-ADE associations of *Ryan et al. (2013)* as the reference set to facilitate comparison.

*Compare results with curated databases:* Cross referencing the LBD output with existing curated databases to verify the validity of the results is another technique. For example, some studies (*Rastegar-Mojarad et al., 2015*; *Cheung, Ouellette & Wasserman, 2012*) have used drug-disease interactions in Comparative Toxicogenomics Database (CTD) to validate their results. Similarly, other databases such as SIDER2 (*Shang et al., 2014*), GEO (*Faro, Giordano & Spampinato, 2011*), GAD (*Seki & Mostafa, 2009*), StringDB (*Nagarajan et al., 2015*) have also been used for validation.

*Compare results using other resources:* In contrast to curated databases, this technique uses other reliable sources such as websites to validate the results. For instance, *Vidal et al. (2014)* have considered the information published in *Mayo Clinic public website* as the ground truth to evaluate the effectiveness of their ranking technique.

### Comparison with baselines:

Different baseline models have been considered for comparison as discussed below.

*Comparison with existing LBD tools:* Several studies have considered the output of the popular LBD tools as the baseline to compare their results. The LBD tools that have been considered for results comparison are; BITOLA (*Lever et al., 2017*), ARROWSMITH (*Loglisci & Ceci, 2011*), Manjal (*Vidal et al., 2014*), ANNI (*Lever et al., 2017*), FACTA+ (*Lever et al., 2017*).

*Comparison with previous LBD techniques:* In this evaluation method, popular techniques that have already been tested by several LBD studies are considered as a baseline to facilitate comparison. This includes techniques like Association Rule Mining (e.g., Apriori *Hu et al., 2010*), distributional semantic techniques (e.g., LSI and RRI (*Hu et al., 2010*)), lexical statistics (e.g., TF-IDF, token frequencies (*Kim et al., 2016*)), and bibliographic coupling (*Sebastian, Siew & Orimaye, 2015*).

*Comparison with previous LBD work:* Several studies have recreated previous LBD methodologies as a baseline to compare their results. Recreating work of *Gordon & Lindsay (1996)* for comparison in *Gordon & Dumais (1998)* and recreating work of *Hristovski et al. (2001)* for comparison in *Huang et al. (2005)* can be considered as examples. Some studies have only recreated subsections of the previous methodologies to evaluate the corresponding sub-section of their methodology. For instance, in *Rastegar-Mojarad et al. (2016)*, they have compared their ranking method with Linking Term count in (*Yetisgen-Yildiz & Pratt, 2006*). Others have straightaway compared the results with the results of the previous methodologies. For example, Qi and Ohsawa *Qi & Ohsawa (2016)* have compared their results of *Migraine ↔Magnesium* rediscovery with five other previous work in terms of precision, recall and F-measure.

*Comparison with other state-of-the-art methods:* Some studies have compared their work with state-of-the-art methods in the relevant disciplines that are not necessarily tested in LBD before. For example, *Crichton et al. (2018)* have considered *Adamic-Adar*, *Common Neighbours* and *Jaccard Index* to compare their results as these algorithms are considered to be competitive and challenging baselines in *link prediction* discipline.

### Expert-oriented evaluation

*Expert-based evaluation:* In expert-based evaluation, typically one (*Gubiani et al., 2017*) or two (*Baek et al., 2017*) domain experts inspect the LBD output to verify if the produced associations are meaningful. Alternatively, the domain expert may provide with a more open-ended evaluation (*Gordon, Lindsay & Fan, 2002*) by asking to provide anticipated future associations in the domain without actually looking at the LBD results. Afterwards, the list of potential associations provided by the expert is cross-checked against the actual LBD outcome. However, expert-based evaluation is expensive, time-consuming and suffers from subjectivity.

*Qualitative analysis of selected results:* A commonly used technique in LBD evaluation is qualitative analysis of the LBD output as an ad-hoc basis by the author(s) or domain expert(s) (*Jha & Jin, 2016a*; *Huang et al., 2016*). Since the complete LBD result is not properly evaluated, it is hard to determine the true accuracy of the LBD methodology. Moreover, same as in expert-based evaluation, the analysis of results suffers from subjectivity.

### User-oriented evaluation

It is crucial to perform user-oriented evaluations to verify the use of the LBD system for real-world usage. However, such evaluations are rarely performed in existing literature.

*User-based evaluation:* Evaluating user's ability to identify and formulating hypotheses from the output of the LBD process is an essential evaluation approach. However, such user-oriented evaluations are mostly neglected in LBD literature. As defined in the study (*Qi & Ohsawa, 2016*), criterions such as, *utility* (how useful is the generated hypothesis is?), *interestingness* (how interesting is the generated hypothesis?) and *feasibility* (to what extent the generated hypothesis can be realised?) can be utilised to score this user formulated hypotheses. These scores can be analysed to verify to what extent the LBD system assist the users to create scientifically sensible novel research hypotheses.

*User-experience evaluation:* Analysing how users interact with the LBD system plays a critical role as such user behaviours provide useful clues to improve the visualisation techniques of LBD results, user-interface, and the process of knowledge discovery. However, user-experience is rarely measured in LBD research. Qi and Ohsawa (*Qi & Ohsawa, 2016*) have compared the performance of experts and non-experts with their matrix-like visualisation LBD process and verified that the users with no prior knowledge also benefit from their LBD process. Similarly, a user performance evaluation was conducted in the study of *Cohen et al. (2010)* using one domain expert and one advanced undergraduate student using a total of nearly 6.5 h of sessions to qualitatively evaluate their LBD tool named as *EpiphaNet* from the user's perspective.

### Proven from experiments

Some studies have experimented the produced hypotheses to prove their validity. Since most of the LBD methodologies are in medical domain, clinical trials are mostly used to verify the derived hypotheses. However, validating all the derived associations of the LBD process using laboratory experiments is infeasible. Hence, the most likely to be successful association from the top of the list is picked to prove (*Baek et al., 2017*). As a result, this evaluation does not assess the accuracy of the remaining associations, thus, does not reflect the overall performance of the LBD methodology.

### Scalability analysis

From query to query, the number of records that need to be analysed vary (*Spangler et al., 2014*). Therefore, it is important to measure how much time and storage required for the different phases of the LBD process to make the methodology more user-friendly.

*Processing time analysis:* Less processing time is a critical characteristic of the LBD process as the users would like to quickly obtain results for their queries. However, the time complexity is rarely measured and compared against other LBD methodologies in the literature. Few LBD studies (*Hossain et al., 2012*; *Loglisci & Ceci, 2011*) have performed processing time analysis of their algorithm.

*Storage analysis:* Analysing memory requirements is important when dealing with large datasets. In the study of *Symonds, Bruza & Sitbon (2014)*, they have analysed the storage complexity of several distributional models. Through their analysis, they have identified that *Tensor Encoding* model is well suited for open discovery as it is efficient in storing and computing independent of the vocabulary size.

### Evaluate ranking technique:

The algorithm used to rank the detected associations plays a vital role in LBD methodology. It should rank the most promising associations in the top of the list by filtering the weak or false-positive associations. Therefore, the success of the LBD process greatly depends on the effectiveness of the ranking algorithm.

*Evaluate ranking positions:* Most of the studies have evaluated the ranking positions of the LBD output to verify the effectiveness of their ranking algorithm. For instance, the LBD studies that have chosen to replicate previous medical discoveries (*Gordon & Dumais, 1998*; *Lindsay, 1999*) have attempted to obtain the associations of that particular medical discovery in the top of the list. Some studies have compared their ranked list with a ranking list of previously published LBD studies to determine the superiority of their algorithm (*Gordon & Dumais, 1998*). Moreover, in techniques such as time-sliced evaluation (*Yetisgen-Yildiz & Pratt, 2009*), the efficiency of the ranking algorithm is measured by using information retrieval metrics such as 11-point average interpolated precision, precision at k, and mean average precision. Some studies have automatically created ground-truths using evidence from literature to evaluate their ranking algorithms (*Xun et al., 2017*).

*Evaluate ranking scores:* Mapping the ranking scores of the detected associations with scores obtained from databases (*Baek et al., 2017*) or other algorithms (*Pusala et al., 2017*) is another evaluation technique used in the literature.

***Evaluate the quality of the output:***

*Evaluate the interestingness of results:* Cameron et al. (2015) have used association rarity to statistically evaluate the interestingness of the LBD output. To facilitate this, they have queried Medline to obtain the number of articles that contain the derived associations and divided it by the number of associations. Afterwards, an interesting score was obtained which is proportionate to the rarity score.

*Evaluation of quality and coherence of stories:* This evaluation metric provides a novel perspective to LBD evaluation. The quality of the produced story chains can be evaluated using dispersion coefficient which is 1 for an ideal story (Hossain et al., 2012). This type of evaluation can be adapted when the LBD methodology outputs a chain of story path (e.g., output of an AnC model).

We also analysed the generalisability of each evaluation technique across domains. To achieve this, the previously discussed evaluation techniques are categorised into the following two groups; *Category1:* Highly domain dependent and only applicable to domains where similar resources are available, and *Category2:* Domain-independent (Table 3).

The most prominent and widely used evaluation technique which is *discovery replication* is only limited to the medical domain. Other popular evaluation techniques such as the *use of curated databases and resources* and *comparison with existing LBD tools* are also highly domain dependent and mostly available for the medical domain. Nevertheless, the most objective evaluation technique considered so far in the discipline, which is *Time-sliced Evaluation* is domain independent. Most of the remaining evaluation techniques are typically independent of the domain and can be utilised in non-medical LBD studies.

### What are the main quantitative measurements used to assess the effectiveness of the results?

Different information retrieval metrics have been used to obtain a quantitative understanding of the performance of the LBD methodologies as summarised in Table 4. From our analysis 3 we observed that *precision* (i.e., fraction of associations obtained from the LBD process that are relevant), *recall* (i.e., fraction of relevant associations that are successfully retrieved), *F-measure* (i.e., harmonic mean of precision and recall) and *Area Under Curve (AUC)* (i.e., area under the Receiver Operating Characteristic (ROC) curve which falls in the range from 1 to 0.5) are the popular metrics used in the literature.

Since most of the time the users will not able to go through the entire list of suggested associations, it is also important to evaluate the proportion of associations in the top k positions that are relevant. For this purpose, the metrics such as *precision at k*, *recall at k*, *11-point average interpolated precision*, and *Mean Reciprocal Rank* have been used in the literature.

## LIMITATIONS

Even though we present the insights gleaned from our rigorous literature analysis with confidence, we may have missed LBD research articles that are outside of the six databases and six keywords we used. To alleviate this issue to some extent, we also included the

**Table 3  Domain dependency of the evaluation techniques.**

| Evaluation Technique | Category1 | Category2 |
|---|:---:|:---:|
| **Evidences-based Evaluation:** | | |
| Replicating existing medical discoveries | ✓ | – |
| Time–sliced evaluation | – | ✓ |
| Manual literature search | – | ✓ |
| Intersection evaluation | – | ✓ |
| Derive reference sets from literature | – | ✓ |
| Compare results with curated databases | ✓ | – |
| Compare results using other resources | ✓ | – |
| **Comparison with baselines:** | | |
| Comparison with existing LBD tools | ✓ | – |
| Comparison with previous LBD techniques | – | ✓ |
| Comparison with previous LBD work | – | ✓ |
| Comparison with other state-of-art methods | – | ✓ |
| **Expert-oriented Evaluation:** | | |
| Expert-based evaluation | – | ✓ |
| Qualitative analysis of several selected results | – | ✓ |
| **User-oriented Evaluation:** | | |
| User-based evaluation | – | ✓ |
| User-experience evaluation | – | ✓ |
| **Proven from Experiments:** | | |
| Clinical Tests (or relevant other experiments) | – | ✓ |
| **Scalability Analysis:** | | |
| Processing time analysis | – | ✓ |
| Storage analysis | – | ✓ |
| **Evaluate Ranking Technique:** | | |
| Evaluate ranking positions | – | ✓ |
| Evaluation ranking scores | – | ✓ |
| **Evaluate the quality of the output:** | | |
| Evaluate the interestingness of results | – | ✓ |
| Evaluation of quality and coherence of stories | – | ✓ |

references from a recent review (*Henry & McInnes, 2017*) during our paper retrieval process as discussed in the *Methods* section.

## DISCUSSIONS AND FUTURE WORK

The key findings and future research directions of each component of the LBD workflow are summarised below.

*Input Component:* The primary source of data utilised in LBD studies is *research papers*. Different studies have extracted different details from the research papers for analysis. Among them, using *title and abstract* is the most popular method. However, some studies have proven the use of full-text, and other metadata such as keywords, references, author details and venue details assist to glean additional clues of the anticipated knowledge links.

**Table 4  Quantitative measures used in the literature.**

| Measure | Past Studies |
| --- | --- |
| Precision | *Lever et al. (2017)*, *Yang et al. (2017)*, *Preiss & Stevenson (2017)* |
| Recall | *Sebastian, Siew & Orimaye (2017b)*, *Jha & Jin (2016a)*, *Sang et al. (2015)* |
| F-Measure | *Preiss, Stevenson & Gaizauskas (2015)*, *Sebastian, Siew & Orimaye (2015)*, *Sang et al. (2015)* |
| Precision at k | *Vlietstra et al. (2017)*, *Shang et al. (2014)*, *Song, Heo & Ding (2015)* |
| Recall at k | *Lever et al. (2017)*, *Vlietstra et al. (2017)*, *Shang et al. (2014)* |
| Average Precision | *Cohen et al. (2012)*, *Roy et al. (2017)* |
| Mean Average Precision | *Yang et al. (2017)*, *Shang et al. (2014)*, *Crichton et al. (2018)* |
| Precision over time | *Yetisgen-Yildiz & Pratt (2006)* |
| Recall over time | *Vlietstra et al. (2017)*, *Yetisgen-Yildiz & Pratt (2006)* |
| 11-point average interpolated precision | *Yetisgen-Yildiz & Pratt (2009)* |
| Area Under Curve | *Lever et al. (2017)*, *Kastrin, Rindflesch & Hristovski (2016)*, *Sebastian, Siew & Orimaye (2015)* |
| Accuracy | *Sebastian, Siew & Orimaye (2017b)*, *Sang et al. (2015)* |
| Cumulative Gain | *Vlietstra et al. (2017)* |
| Mean Reciprocal Rank | *Song, Heo & Ding (2015)* |
| Correlation Analysis | *Baek et al. (2017)*, *Yang et al. (2017)*, *Xun et al. (2017)* |

*Lee et al. (2015)* point out that *different perspectives* are reflected by different data types used in the content of the research papers. In their analysis, they have found that *keyphrases*, *citation relationships*, and *MeSH* reflect the views of *authors*, *citers*, and *indexers* respectively. Moreover, *Kostoff et al. (2004)* have analysed the *information content* in various fields of a paper using four metrics; total number of phrases, number of unique phrases, factor matrix filtering and multi-link hierarchical clustering. They have identified that the selection of the field depends on the objectives of the study as described in *Kostoff et al. (2004)*. Hence, selecting the suitable data type in the papers in crucial as they represent different *perspectives* (*Lee et al., 2015*) and *information content* (*Kostoff et al., 2004*) and mainly depends on the objective of the research. Furthermore, *Nagarajan et al. (2015)* have discovered that the LBD performance mainly depends on the richness of the information being used.

Apart from research papers, several approaches have experimented the LBD process with other traditional data types such as patents and clinical case reports. *Smalheiser, Shao & Philip (2015)* have identified that *information nuggets* (i.e., main findings) are surprisingly prevalent and large in clinical case reports. Mostly, the title itself reveals the main findings of the case report that enable ample opportunity for *finding-based information retrieval* (*Smalheiser, Shao & Philip, 2015*).

Interestingly, LBD methodology was successfully adopted to non-traditional data types such as drug labels, tweets, news articles, and web content. Therefore, an interesting future direction would be to analyse how the LBD process using research papers can be enhanced by integrating knowledge from non-traditional data types such as tweets. Furthermore,

since most of the non-traditional data types are utilised in medical domain another interesting avenue would be integrate LBD process in other domains using data types such as product descriptions (for product recommendation), movie scripts (for movie recommendation), and recipe books (for recipe recommendation).

With respect to unit of analysis, making use of controlled vocabularies such as UMLS, MeSH, Entrez Gene to extract concepts is the most popular approach. However, research outside of medical domain have followed a term-based approach by extracting n-grams. As the controlled vocabularies utilised yet in LBD research are in the medical domain, an interesting future avenue is to experiment the use of general-purpose controlled vocabularies (such as DBpedia, Freebase, and YAGO) to facilitate knowledge discovery in a cross-disciplinary manner.

*Process Component:* Swanson's manually detected medical discoveries have set the base for the LBD research. Later various computational techniques such as statistical, knowledge-based, relations-based, hierarchical, graph-based, bibliometrics-based, link prediction etc. were proposed to automate the process of LBD. The *filtering* and *ranking* techniques used in the LBD methodology are two equally important major components of the LBD workflow.

Many of the filtering mechanisms utilised in LBD studies have restricted the search space using *word-level* filters. Considering the *article-level* filters (e.g., analysing the contribution of outlier documents), *section-level* filters (e.g., analysing the contribution of different sections in a research article such as introduction and conclusion), or *sentence-level* filters (e.g., analysing the contribution of sentences that describes the main findings) have received little attention in the literature. Therefore, analysing the effect of various *article*, *section* and *sentence* level filtering techniques to remove noisy associations before the word-level filtering is another important area that needs to be further explored. Ultimately, such techniques will also help to further narrow down the literature search and to eliminate the hindrances of the existing word-level filters.

As for the ranking techniques, most of the studies have utilised conventional statistical measures to rank/threshold their results. Whether using such single measure alone would be sufficient to rank the most promising associations in the top of the list is doubtful. In other words, an association may require satisfying several characteristics to become a significant and promising association among others. Therefore, it would be more interesting to develop a ranking approach that reflects the identified characteristics of potential associations to prioritise the results. For instance, *Torvik & Smalheiser (2007)* have attempted to derive a formula using seven features that capture various characteristics of an association into a single score by employing a machine learning model. Identifying the important characteristics of a significant and promising association and deriving a score based on these characteristics to rank the LBD results would be more successful than merely relying on standard single measures. In this regard, the analysis of different types of gaps in the literature is useful (*Peng, Bonifield & Smalheiser, 2017*). Moreover, *Smalheiser (2017)* suggests the need of several ranking measures to customise the LBD output according to the user preferences. LION LBD system (*Pyysalo et al., 2018*) that supports multiple scoring functions to facilitate flexible ranking mechanism can be taken as an example.

*Output component:* The typical output of the LBD process is a ranked list of terms that denote the potential associations. However, it is not an effective output technique as the users need to interpret the logical connections of the associations by manually reading the research articles which is difficult and time-consuming. As a result, other visualisation techniques such as term groupings, graphs, and discovery pathways have been proposed in LBD literature. However, the extent to which these proposed techniques assist the user has been rarely measured. Therefore, providing a better visualisation (which is concise, easily interpretable, less complex, visually attractive, and assist users to gain new knowledge) and measuring the user experience of the visualisation are two critical components of LBD workflow that need to be further explored by integrating HCI techniques.

Nevertheless, the importance of such techniques has been overlooked by the LBD community. To date, only a few LBD research (*Wilkowski et al., 2011*; *Hristovski et al., 2006*) have contributed in terms of user interaction studies. These studies make use of *Information Foraging Theory* which is a technique that analyses the user's information retrieval behaviour. The theory evaluates the user's information seeking behaviour in terms of *costs* and *benefits*. If the user can maximise his/her rate of gaining valuable information (i.e., *maximum benefit*) by spending the lowest amount of energy (i.e., *minimum effort*), it is called as an optimal foraging. The key concepts in an information-seeking context are *information*, *information patches*, *information scents* and *information diet* which needed to be supported effectively when designing interfaces (*Ruthven & Kelly, 2011*). Therefore, the challenge of information visualisation is to discover effective mechanisms to represent massive amounts of data and provide effective ways to navigate through them to support users with optimal foraging. The novel advances in HCI research will be useful in this regard (*Stephanidis et al., 2019*). Moreover, Smalheiser and Torvik *Smalheiser & Torvik (2008)* emphasises the importance of simplicity in user-interfaces of LBD tools to support widening the target audience.

*Evaluation component:*

Evaluating the LBD output is challenging and remains to be an open issue as the field lacks gold standard datasets or consistent formal evaluation techniques. The most widely used evaluation technique is replicating Swanson's medical discoveries. However, relying merely on discovery replication can be restrictive and may fail to reflect the true performance of the LBD methodology. Hence, this technique should be accompanied with other evaluation techniques to overcome these limitations. Another popular technique is qualitatively evaluating the results randomly by an expert or author. Nevertheless, this does not give an overall image of the LBD system's performance as few valid associations are taken into consideration for discussion. An LBD system that produces a handful of valid associations in a sea of invalid associations tend to be inefficient (*Yetisgen-Yildiz & Pratt, 2009*). As a result, besides this random quantitative evaluation, the system should also be validated qualitatively to measure the overall performance of the system.

To date, *time sliced evaluation* is considered as the most objective evaluation technique proposed in the field. However, this evaluation technique suffers from two major limitations; (1) The association is proven to valid if the starting and linking term co-occur in future publications (that do not co-occur in the training set). However, co-occurrence

does not necessarily mean that the proposed link has been established, and (2) Rejected associations can still be valid even though they have not been published yet.

To overcome the first limitation, it is important to perform much deeper analysis of language (*Korhonen et al., 2014*) to verify whether the co-occurrence imply a true association which can be considered as an interesting future direction. Additionally, some studies have attempted to utilise evidences from *curated databases* (e.g., CTD, StringDB) as an alternative for co-occurrence in time-sliced evaluation. However, such curated databases are limited to certain problems and may not be available for every domain or problem. The second limitation of time-sliced evaluation can be alleviated to some extent through domain expert involvement by further evaluating the rejected associations.

Another interesting direction for future evaluation is to incorporate the actual end users of LBD research to validate the results which is a neglected area in the literature. For instance, involving users with a diverse range of knowledge and expertise (e.g., novice to expert) will help to understand the extent to which each user will be benefited from the LBD output. In this regard, the hypotheses scoring mechanism used by *Qi & Ohsawa (2016)* can be considered as a successful first step.

Due to the massive influx of scientific knowledge, the volume of data that the LBD system expects to analyse increases with time. For instance, a simple search of "*dementia*" results in more than 150,000 records in PubMed alone. This highlights the importance of performing scalability analysis of the LBD systems in terms of time and storage. This will also improve the usability of the system.

## CONCLUSION

In this review, we present novel, up-to-date and comprehensive categorisations to answer each of our research questions to provide a detailed overview of the discipline. The review summary and a comparison with the following recent reviews (*Henry & McInnes, 2017*; *Gopalakrishnan et al., 2019*) are available at https://tinyurl.com/workflow-summary.

With respect to *input* component, it is evident that LBD community is showing a growing research interest towards integrating knowledge from non-traditional data sources to enhance the traditional setting of the LBD framework and to explore new application areas. Nevertheless, the selection of the input needs to be precise and cross-checked against the research objectives as different data types reflect different *perspectives* (*Lee et al., 2015*) and *information content* (*Kostoff et al., 2004*). Filtering and ranking are two important constituents of the *process* component. Most of the filtering techniques examined in the discipline are at word-level. However, the importance of article-level, section-level and sentence-level filters have been rarely studied in the literature. Considering the ranking component, most of the studies have employed a single conventional ranking technique to prioritise the generated discoveries. This showcase the need of developing a series of interestingness measures that customise the LBD output that suit multiple scientific investigations (*Smalheiser, 2012*).

The *output* component of LBD workflow, is largely neglected in the prevailing literature which emphasises the necessity of conducting user-interaction studies to assess the user

experience. Concerning the *evaluation* component, time-sliced evaluation is the current most objective technique used to validate the results. However, this technique suffers from several limitations which suggests the requirement of developing new evaluation methods and metrics to evaluate the generated output.

We hope that the future LBD studies will contribute to overcome the prevailing research deficiencies in the LBD workflow with the ultimate intention of uplifting the typical research procedures which are followed by the scientists.

### Funding
The authors received no funding for this work.

### Competing Interests
The authors declare there are no competing interests.

### Author Contributions
- Menasha Thilakaratne conceived and designed the experiments, performed the experiments, analyzed the data, contributed reagents/materials/analysis tools, prepared figures and/or tables, performed the computation work, authored or reviewed drafts of the paper, approved the final draft.
- Katrina Falkner and Thushari Atapattu conceived and designed the experiments, prepared figures and/or tables, authored or reviewed drafts of the paper, approved the final draft.

### Data Availability
This is a review article; there is no raw data.

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
