# Peer review of "A systematic review on literature-based discovery workflow"

_PeerJ Computer Science, doi:10.7717/peerj-cs.235_

## Round 0.1 · original submission · Major Revisions

Please refer to the reviewers’ opinions to improve your paper. Please also write a revision note such that the reviewers can easily check whether their comments are fully addressed. We look forward to receiving your revised manuscript soon.

·

Basic reporting

This review is clearly written, and considers many technical aspects of literature based discovery in a very systematic and easy-to-read manner.

Experimental design

The study design is a major problem. The intent of the authors is to make a systematic review, but the queries that they entered are too specific and woefully inadequate to find all relevant articles. They missed some articles by Swanson himself (e.g., Swanson, 2011) and missed five articles by me that were written after 2007. I did not attempt to find all relevant articles that they missed, but I believe they also missed many articles including by Kostoff, Trevor Cohen, Rindflesch, Spangler, and so on.

Validity of the findings

Descriptively, the text appears to be valid as a snapshot of a certain type of literature based discovery. However, the text does not discuss much about emerging models not based on ABC (Smalheiser, 2012) or related issues such as identifying and resurrecting neglected lines of research. So, although it is nice in organizing a description of many ABC studies, it has limited value in pointing the way towards the future.

·

Basic reporting

The manuscript is well-written and highly readable. Literature-based discovery (LBD) inherently addresses multidiciplinary topics, hence the current review is clearly within the scope of this journal.

The manuscript provides a valuable perspective toward LBD and can certainly benefit researchers in the field. However, there are several aspects of the manuscript that the authors should consider revising before being recommended for a full acceptance. To aid with editorial decision, I have categorised these aspects into "Highly Important" and "Important" categories:

Highly Important

1.1 Notwithstanding the comprehensiveness of this review, there have at least been 3 significant LBD reviews published since 2017: Sebastian et al. and Henry and McInnes in 2017, and Gopalakrishnan et al. in 2019. In view of these competing reviews, the authors is advised to put forward a stronger case to justify the contribution of the current review to the domain. Firstly, the authors should consider providing an additional comparison against the most recent review by Gopalakhrisnan et al. (2019) (https://www.sciencedirect.com/science/article/pii/S1532046419300590). Secondly, more robust discussions on the differentiating feature of the current review against the 3 preceding review papers should be included in order to strengthen the manuscript's contribution. Currently, such discussion is only limited to one paragraph (line 81 - 91) which is not sufficient.

1.2 Readers should expect a review paper to offer valuable expert insights into the LBD field. As such, it is not enough to merely present highly organised facts about LBD methods and systems (which the current manuscript has done very well). The authors are encouraged to provide more insightful commentaries of pertinent research and development gaps of LBD. There is an attempt to address these in the "Conclusion and Future Work" section, but similar commentaries have been found in other recent review papers mentioned above. For instance, the authors may consider dwell a little more of the issues of usability of LBD systems (i.e. HCI) which is not much talked about elsewhere.

1.3 The authors should also emphasise on reviewing the issues surrounding LBD's discovery models. The ABC model should not be taken for granted and ought to be scrutinised. Considering drawing conceptual links with three important LBD commentaries: Davies (1989) https://www.emerald.com/insight/content/doi/10.1108/eb026846/full/html, Smalheiser (2012) https://onlinelibrary.wiley.com/doi/full/10.1002/asi.21599 , and Smalheiser (2017) https://www.ncbi.nlm.nih.gov/pmc/articles/PMC5771422/ (note that all 3 papers are not present in the current review).

Important

1.4 Content in line 717 - 753 is a duplication of the content in line 682 - 716.

1.5 Consider adding graphics (as appropriate) to improve readibility of the manuscript.

1.6 Paraphrase the quote in line 46 - 50.

1.7 Kindly check for and fix grammatical/punctuation errors, such as in line 78 - 80, 144, and 176.

1.8 "heterogeneous bibliographic information" should be correctly stated as "heterogeneous bibliographic information network" (line 74).

Experimental design

The overall study design is sound. The review is organized logically into coherent paragraphs/subsections.

2.1 One of the most obvious strength of the current review is its systematic review approach. To my knowledge, no other LBD review has attempted to adopt such rigorous study design. Having said that, the authors should further elaborate on these procedure to strengthen their contribution.

2.2 Another strength of this manuscript is its additional emphasis on reviewing LBD workflow. Including more reviews on how LBD systems has been used in various real world scientific investigation scenarios is essential.

2.3 Sources are adequately cited, although several recent LBD methods/systems might have been overlooked by the authors, e.g:
https://journals.plos.org/plosone/article?id=10.1371/journal.pone.0215313
https://academic.oup.com/bioinformatics/article/35/9/1553/5124276

Consider including additional articles above in the review.

Validity of the findings

3.1 On page 3, a little more discussion is needed to justify the selection of the 8 research questions. This is important as the rest of the review hinges on these questions. Also, consider removing "RQ" from the rest of the manuscript to avoid the impression that the manuscript is taken straight from a postgraduate thesis.

3.2 Consider moving the discussions currently located in the "Conclusion and Future Work" section to a dedicated section called "Discussions and Future Work". Keep the conclusion brief.

Additional comments

Providing detailed lists of various aspects in LBD workflow is certainly a strength of the authors' manuscript. They may serve as a useful, easy-to-refer guide for future LBD researchers/developers.

On the other hand, to publish a new LBD review in the span of 3 years during which 3 reviews have been previously published certainly demand justifications to minimise redundant contributions.

Assuming that the authors address the various feedback above, I believe that the current manuscript will have two important contributions to make to LBD: 1) providing the first systematic review of the field, and 2) providing a fresh look at LBD from the workflow and usability perspectives.

---

## Round 0.2 · accepted · Accept

I have received the review reports for your paper submitted to PeeJ from the reviewers. According to the reports, I will recommend acceptance to your paper.

·

Basic reporting

No comment

Experimental design

No comment

Validity of the findings

No comment

Additional comments

The coverage of the review is much improved.

·

Basic reporting

No comment. The authors have comprehensively addressed the previously suggested improvements.

Experimental design

No comment. The authors have comprehensively addressed the previously suggested improvements.

Validity of the findings

No comment. The authors have comprehensively addressed the previously suggested improvements.

Additional comments

You have rigorously addressed my previous suggestions for improvement. I believe the comprehensiveness and the fresh perspectives offered by this paper represent a significant contribution to the expanding field of literature-based discovery.